# OnabotulinumtoxinA Modulates Visual Cortical Excitability in Chronic Migraine: Effects of 12-Week Treatment

**DOI:** 10.3390/toxins15010023

**Published:** 2022-12-29

**Authors:** Angelo Torrente, Laura Pilati, Salvatore Di Marco, Simona Maccora, Paolo Alonge, Lavinia Vassallo, Antonino Lupica, Serena Coppola, Cecilia Camarda, Nadia Bolognini, Filippo Brighina

**Affiliations:** 1Department of Biomedicine, Neurosciences and Advanced Diagnostics, University of Palermo, 90127 Palermo, Italy; 2Headache Center “Casa della Salute Cittadella San Rocco”, AUSL Ferrara, 44121 Ferrara, Italy; 3Neurology Unit, ARNAS Civico di Cristina and Benfratelli Hospitals, 90127 Palermo, Italy; 4Department of Psychology & Milan Center for Neuroscience—NeuroMi, University of Milano Bicocca, 20126 Milano, Italy; 5Laboratory of Neuropsychology, IRCSS Istituto Auxologico Italiano, 20122 Milano, Italy

**Keywords:** onabotulinumtoxinA, chronic migraine, multisensory integration, visual cortical excitability, neurophysiology

## Abstract

Chronic migraine is a burdensome disease presenting with episodic pain and several symptoms that may persist even among headache attacks. Multisensory integration is modified in migraine, as assessed by the level of the perception of sound-induced flash illusions, a simple paradigm reflecting changes in cortical excitability which reveals to be altered in migraineurs. OnabotulinumtoxinA is an effective preventive therapy for chronic migraineurs, reducing peripheral and central sensitization, and may influence cortical excitability. Patients affected by chronic migraine who started onabotulinumtoxinA preventive therapy were included. Clinical effects (headache diaries and migraine related questionnaires) were assessed at the beginning of the therapy and after 12 weeks. Contextually, patients underwent the evaluation of multisensory perception by means of the sound-induced flash illusions. OnabotulinumtoxinA showed effectiveness both in migraine prevention and in reducing headache burden. Even one session of therapy was able to restore, at least partially, multisensory processing, as shown by patients’ susceptibility to the sound-induced flash illusion. OnabotulinumtoxinA could influence migraineurs cortical excitability concurrently to the beneficial effects in headache prevention.

## 1. Introduction

Migraine is one of the most prevalent neurological diseases and it is estimated that 14.4% of the global population suffers from it, representing a serious social issue. A recent analysis showed how migraine represents the second cause of years lived with disability after low back pain; however, if we consider only the population in the working age groups (second, third, fourth, and fifth decades of life), migraine reaches the first place [1]. Migraine may present with a variable frequency among different subjects and, even in the same subject, among different life epochs. The International Classification of Headache Disorders (ICHD-3) defines migraine as “chronic” when a patient complains of 15 or more days of headache per month (at least eight of which show migraine characteristics) for more than 3 months [2]. Indeed, chronic migraine constitutes a very burdening disease that limits patients’ lives [3]. Even if pain may not be present daily, several other disabling symptoms can occur. For instance, chronic migraineurs suffer from attention and memory impairments, as well as constant phono- or photophobia. Such symptoms depend on chronic cortical changes (i.e., hyperexcitability) and brainstem alterations (i.e., periaqueductal gray involvement) that lead to an impaired processing of external stimuli and to an altered pain modulation [4].

Migraine episodes can be triggered by visual, auditory, or olfactive stimuli; furthermore, migraineurs show a hypersensitivity to environmental stimulation that manifests as phonophobia, photophobia, or osmophobia or cutaneous allodynia; these phenomena seem to be related to one another, as the exposure to stimuli of one modality (e.g., light) enhances even the sensitivity of other sensory modalities too (e.g., touch) [5]. These findings suggest an involvement of the mechanisms related to processing and integration of inputs from different sensory modalities. Therefore, multisensory integration represents an interesting mechanism to study in migraineurs. When different stimuli come from the same source from different sensory modalities, our brain tends to integrate them into a unique percept (e.g., lipreading allows one to better understand spoken words). Sometimes, this occurs when sensory stimuli are incongruent, and their interaction may cause perceptual illusions. An example of this is the sound-induced flash illusion (SIFI), which has been used to assess multisensory processing in migraine sufferers and its link to migraine pathophysiology. In the SIFI, when a visual stimulus (e.g., a white dot, i.e., a *flash*) is presented together with two or more auditory stimuli (i.e., *beeps*), a subject tends to perceive more than one flash (*fission* illusion); conversely, when multiple flashes are presented with just one beep, the subject tends to perceive less flashes than presented (*fusion* illusion) [6,7]. The SIFI is a very simple paradigm that can be administered by using computer software. Interestingly, it has been shown that fission and fusion effects are associated to changes in temporal and occipital cortical excitability; in fact, it is possible to modulate such illusions via transcranial direct current stimulation (tDCS) [8]. There is strong evidence that the SIFI is altered in migraine patients, suggesting that multisensory integration is altered in migraine. Particularly, episodic migraineurs show increased visual cortical excitability compared to healthy subjects, which is likely responsible for a reduced susceptibility to SIFI, both during migraine attacks and interictally [9]. In chronic migraine, the disruption of multisensory processing, as indexed by SIFI, is greater especially under triptan overuse [10]. Migraineurs’ cortical excitability can be modulated with pharmacological therapy, as demonstrated by studies with transcranial magnetic stimulation (TMS) in patients treated with valproate or topiramate [11,12].

One of the main pathophysiological mechanisms of migraine chronification is represented by the sensitization of nociceptive pathways. It starts at peripheral nociceptors, particularly the ones in trigeminal innervated skin, and then moves to second-order neurons at the trigeminal nucleus (central sensitization) [13]. The last phenomenon is responsible for cutaneous allodynia that may be present during a migraine attack or even interictally. Moreover, a chronically persisting nociceptive activation is believed to cause the sensitization of higher central structures (e.g., thalamus), leading to a vicious circle that further worsens migraine [13]. The onabotulinumtoxinA or botulinum neurotoxin A (BoNT-A) is an approved therapy for resistant chronic migraine, with quarterly injections of 155–195 UI on 31–39 target muscles [14,15]. Preclinical studies showed how BoNT-A is able to reverse C-meningeal nociceptors’ sensitization, as well as to reduce the release of inflammatory mediators in the trigeminal ganglion [16,17] and calcitonin gene-related peptide (CGRP) plasma levels in chronic migraineurs [18]. Accordingly, BoNT-A is believed to be able to affect peripheral and central sensitization in chronic migraine [19].

The aim of the present study is to investigate whether BoNT-A preventive therapy in chronic migraineurs is able to modulate multisensory integration as measured by means of the SIFI.

## 2. Results

### 2.1. Population

Sixteen patients (all females, mean age = 53.6 ± 10.8 years) affected by chronic migraine who started preventive therapy with BoNT-A participated in the study providing their written informed consent.

### 2.2. Clinical Findings

#### 2.2.1. Headache Characteristics and Medications

Patients reported a significant reduction in Mean Monthly Headache Days (MMHDs) after 12 weeks of BoNT-A therapy (t0 = 26.2 ± 4.7 days vs. t1 = 16.9 ± 5.6, *p* < 0.001), showing a mean reduction of 35.6 ± 18.8% days. The mean intensity of headache, as measured using a Visual Analogue Scale (VAS), significantly reduced at the end of the therapy (t0 = 8.5 ± 0.9 vs. t1 = 7.7 ± 0.9, *p* = 0.022). Regarding the response to therapy, as measured by the percentage of MMHDs’ reduction (see Section 5), seven (43.8%) patients were classified as non-responders, nine (56.3%) were partial responders, and five were (31.3%) responders after 12 weeks of therapy. Lastly, patients did not complain about any BoNT-A therapy side effect.

Regarding the use of acute medications, eight (50.0%) patients usually used triptans, six (37.5%) used nonsteroidal anti-inflammatory drugs (NSAIDs), one (6.25%) used acetaminophen, and one (6.25%) used corticosteroids. Furthermore, following ICHD-3 definition [2], fifteen (93.4%) patients overused medications at t0, while, at t1, the number reduced to ten (62.5%). As far as the Mean number of Days with (acute) Medications (MDMs) is concerned, patients reported a significant reduction in acute drug intake (t0 = 23.9 ± 6.5 vs. t1 = 16.1 ± 9.0, *p* = 0.001).

#### 2.2.2. Headache Impact and Disability

Patients showed a significant reduction in the impact of headache in everyday life, as well as the related disability: Headache Impact Test-6 (HIT-6) scores reduced from 67.8 ± 5.3 at t0 to 64.4 ± 7.4 at t1 (*p* = 0.018); Migraine Disability Assessment (MIDAS) scores reduced from 108.9 ± 85.7 at t0 to 73.2 ± 62.2 at t1 (*p* = 0.004).

Clinical findings are summarized in Table 1 and are shown graphically in Figure 1.

### 2.3. Sound-Induced Flash Illusion

Patients’ performance (mean number of seeing flashes) of the fission illusion analyzed with repeated measures Analysis of Variance (rmANOVA) showed a significant main effect of time × condition (F_4, 60_ = 2.80, *p* = 0.034). The graphic illustration of these changes is showed in Figure 2 and in Table 2. Post-hoc comparisons (Duncan test) showed a significant variation in 1F3B (t0 = 1.8 ± 0.5 vs. t1 = 2.1 ± 0.4, *p* = 0.011) and in 1F4B (t0 = 1.9 ± 0.7 vs. t1 = 2.3 ± 0.5, *p* < 0.001) conditions only. In contrast, the analysis of the fusion illusion did not show any significant main effect (F_5, 75_ = 2.80, *p* = 0.512).

## 3. Discussion

OnabotulinumtoxinA therapy following the phase III research evaluating migraine prophylaxis therapy (PREEMPT) (follow-the-pain) protocol confirmed to be an effective preventive therapy for resistant chronic migraine, as demonstrated by several studies in the literature [14,20,21], even in the long term [22,23]. In the present study, just after 12 weeks of treatment, a good portion of patients obtained a significant reduction in MMHDs (56.3% partial responders and 31.3% responders), a rate expected to increase with repeated sessions of therapy. Not only did headache frequency decrease, so too did its intensity, with pain becoming more tolerable at the end of the therapy. In addition, even the reduction in acute drug intake is a remarkable result, both for patients’ health (i.e., less acute drugs side effects) and cost. Moreover, the BoNT-A therapy effects do not limit themselves just to headache prevention: they are even able to reduce the headache impact on patients’ lives [15]. In the present study, a significant reduction emerged in both HIT-6 and MIDAS scores even after just one therapy session, demonstrating how BoNT-A therapy reduces migraine burden.

Regarding therapy effects on multisensory processing, as measured with a SIFI paradigm [6], these illusions are of interest because they imply cortical excitability shifts [7,8]. Previous studies have demonstrated how migraineurs show less fission effects compared to healthy subjects [9]. This phenomenon is more evident in chronic migraine, in which headache is more frequent and the cortex may not return to interictal conditions. Moreover, chronic migraineurs often present medication overuse, and a relation between triptan overuse and SIFI reduction (suggesting that triptan overuse may enhance cortical excitability) has been demonstrated [10]. Since pharmacological therapy is able to modify cortical excitability [11,12], the aim of this study was to evaluate the effects of BoNT-A therapy on multisensory processing. The fission illusion significantly increased in some conditions (when 3 or 4 beeps were combined with a single flash, i.e., 1F3B and 1F4B conditions) after the 12 weeks of therapy, while there was no change in the fusion illusion. The fission phenomenon of the SIFI is associated to changes in cortical excitability, while the fusion illusion is less influenced by cortical excitability changes and may depend more on individual factors [8]. Therefore, the partial restoration of fission effects after the OnabotulinumtoxinA therapy suggests a potential effective modulation of visual cortical excitability. Since BoNT-A mainly acts at the level of peripheral sensitization, its influence on cortical excitability suggests secondary central effects [17,19] likely responsible for changes in multisensory processing.

This study presents some limitations. First, a control group is missing; therefore, it is not possible to investigate any placebo effect. Second, concerning the small sample size, future studies are needed to confirm the present findings in a larger sample to reduce the interindividual variability effect. Moreover, the present effects need to be confirmed in the long term, comparing data of the same patients even after 24, 36, and 48 weeks of BoNT-A therapy. Furthermore, it could be of interest to study how long the effects will endure in patients that discontinue the therapy (for efficacy or inefficacy). Given the observed multisensory effects induced by the BoNT-A therapy, it could be interesting to evaluate even the impact on sensory symptoms (e.g., photophobia or phonophobia). Lastly, another interesting comparison of SIFI could be performed with other preventive therapies used in chronic migraine, such as monoclonal antibodies against CGRP.

## 4. Conclusions

The present work demonstrates the efficacy of a 12-week BoNT-A preventive therapy on chronic migraine. The benefits reach beyond a reduction in headache frequency, improving also patients’ quality of life and reducing the consumption of acute drugs. Moreover, a positive modulation of multisensory perception was demonstrated, suggesting changes in visual cortical excitability.

## 5. Materials and Methods

### 5.1. Protocol Approval and Informed Consents

The study protocol was approved from the Palermo 1 Ethical committee (Palermo, Italy) and it was written following the principles of the Declaration of Helsinki [24]. Investigators obtained patients’ written informed consent before their inclusion in the study.

### 5.2. Participants

Adult patients suffering from chronic migraine (diagnosed from neurologists expert in the field following ICHD-3 criteria [2]) for whom two or more preventive treatments were ineffective or contraindicated and who started a preventive therapy with BoNT-A (as suggested by recent guidelines [25]) were included. At the time of the inclusion, patients must not have started other migraine preventive therapies or, if present, their dosage must not have been modified within the previous 3 months. The exclusion criteria consisted of photosensible epilepsy, severe hypoacusis, or a severe reduction in vision acuity, scarce compliance, or difficulties in performing requested tasks (i.e., dementia, attention deficit, impossibility to maintain sitting position).

### 5.3. Study Design

The present work is a prospective observational study, evaluating the neurophysiological effects of BoNT-A preventive therapy on chronic migraine. Since the effectiveness of the therapy has already been demonstrated, authors did not include any control population. Moreover, during every session, patients’ clinical findings were collected, even using auto-administrable questionnaires.

The evaluations were performed on the day of the first BoNT-A injection session (t0) and after 12 weeks during the second session (t1). To avoid any influence related to the pain of the injections, patients underwent visual cortical excitability and clinical assessment first and the injective therapy after.

### 5.4. Preventive Treatment Protocol

Patients underwent therapy with 195 UI of BoNT-A every 12 weeks, using the PREEMPT follow-the-pain protocol [14]. The injections were performed by a trained neurologist using disposable 1 mL syringes with a 29 G needle, administering 5 UI (0.1 mL) across 39 standardized sites: 5UI at the procerus muscle site, 10 UI across the two sites of the corrugator muscles, 20 UI across the four sites of the frontalis muscles, 50 UI across the ten sites of the temporalis muscles, 40 UI across the eight sites of the occipitalis muscles, 20 UI across the four sites of the cervical paraspinalis muscles, and 50 UI across the ten sites of the horizontal trapezii muscles. Any immediate side effect was investigated in the minutes after the therapy, while late ones were asked during the following session. Furthermore, during the treatment, other potential preventive therapies should not have been modified.

### 5.5. Clinical Findings

#### 5.5.1. Headache Characteristics and Medications

Headache frequency was investigated using paper headache diaries that were given to patients during previous outpatient visits. For each examination (i.e., t0 and t1), the MMHDs of the previous 3 months were recorded. To evaluate the therapy response, three categories of patients were distinguished based on the % reduction in MMHDs: *non-responders* (mean reduction < 30%), *partial responders* (mean reduction ≥ 30%), and *responders* (mean reduction ≥ 50%). Moreover, patients had to indicate on a VAS the mean intensity of their headache. Lastly, patients were asked about the MDMs of the previous 3 months (i.e., the number of days in which the patient took any acute medication for headache) and to tell operators the type of medication used (e.g., triptans, nonsteroidal anti-inflammatory drugs, corticosteroids, acetaminophen, opioids).

#### 5.5.2. Headache Impact and Disability

Migraine impact on patients’ everyday life was investigated, analyzing the scores of two self-administrable questionnaires, the Headache Impact Test-6, and the Migraine Disability Assessment.

##### Headache Impact Test-6 (HIT-6)

HIT-6 is a six-item test designed to assess the headache impact in everyday life, revealing itself to be particularly useful for migraineurs screening or monitoring during follow-ups [26]. HIT-6 evaluates the severity of a headache, its negative effect on daily living, its physical, work, or social limiting effect, and the psychological distress associated to it. The six items are constituted by direct questions about the negative effect of a headache on the patient’s life, and he/she is asked to answer with the frequency of such an effect. The frequencies are *never* (6 points), *rarely* (8 points), *sometimes* (10 points), *very often* (11 points), and *always* (13 points); score range is 36–78, and higher scores indicate a greater impact on the patient’s life.

##### Migraine Disability Assessment (MIDAS)

MIDAS is a simple 5-item questionnaire designed for migraine patients; it is a useful and reliable tool used to assess a headache-associated disability during the 3 months prior to the investigation [27]. It consists of five questions aiming to quantify the total number of days lived with disability due to headache; particularly, they investigate paid or schoolwork (days off and days when productivity is reduced by 50% or more), then household work (in the same way); furthermore, the last question evaluates the number of missed days of recreational or social activities. The total score is obtained with the sum of the number of days reported, and it is divided into 4 disability grades: I (0–5) *minimal* or infrequent disability; II (6–10) *mild* disability; III (11–20) *moderate* disability; IV (more than 20) *severe* disability in everyday life.

### 5.6. Sound-Induced Flash Illusions

Sound-induced flash illusions (SIFI) were evaluated with the stimuli and procedures already described in previous studies of the same research group [8,9,10]. Stimuli presentation (i.e., type, number, timing, and localization) and patients’ responses were set and recorded using computer software (E-prime version 2.0^®^, Psychology Software Tools, Pittsburgh, PA, USA); sound intensity was checked with a decibel meter for a personal computer, and visual stimuli luminance was evaluated with a photometer (Konica Minolta CS-100 Spot Chroma Meter; Konica Minolta, Tokyo, Japan).

Participants were evaluated in a dimly illuminated room, sitting approximately 57 cm from a cathode ray tube (CRT) computer monitor (Samsung SyncMaster 1200NF, Samsung Electronics Italia S.p.A, Milano, Italy: resolution 1024 × 768, refresh rate 75 Hz), with their eyes pointing at the center of the screen (indicated with a white cross). Two speakers were located on the two sides of the screen at the same height of visual stimuli. Each trial began with the appearance of the white fixation cross (luminance = 0.02 cd/m^2^); then, a white disk (luminance 118 cd/m^2^) subtending 2° of visual field was flashed 1 to 4 times (i.e., flash [F]) in a position with an eccentricity of 5° on the low part of the screen. Furthermore, auditory stimuli (i.e., beeps [B]) with an intensity of 80 dB and frequency of 3.5 kHz were administered 0 to 4 times with flashes. Each flash and beep lasted one screen refresh (13 ms); a stimulus onset asynchrony of 5 refreshes (65 ms) was set between flashes, and of 4 refreshes (52 ms) between beeps; the first flash appeared 26 ms after the onset of the first beep (see Figure 3). The task included 11 experimental conditions: (i) *single flash trials* (1F), accompanied by 0 to 4 beeps (B) (i.e., 1F0B, 1F1B, 1F2B, 1F3B, 1F4B) to induce the “fission” illusions and (ii) *multiple flash trials* (i.e., 2F, 3F, 4F), accompanied by 0 or 1 beep (2F0B, 3F0B, 4F0B, 2F1B, 3F1B, 4F1B) to induce the “fusion” illusions. Subjects were asked to report the number of flashes perceived, which was noted by an operator. Every patient was first trained with 10 practice trials (not included in the analysis) and then 8 trials for each condition were presented in a random order (total = 88 trials) for a total duration of approximately 5 min.

### 5.7. Statistical Analysis and Data Presentation

Continuous variables were reported as mean ± standard deviation (SD). Statistical analyses were performed using Statistica Software (Statsoft, Version 7.0, StatSoft Italia SRL, Vigonza, PD, Italy). To assess the fission illusion, we used a repeated measures Analysis of Variance (rmANOVA) test, investigating two within-subject factors: “time” with 2 levels (t0 and t1) and “condition” with 5 levels (1F0B, 1F1B, 1F2B, 1F3B, 1F4B). A post-hoc Duncan test was performed to investigate which conditions varied significantly. To study the fusion illusion, another rmANOVA test was conducted, investigating two within-subject factors: “time” with 2 levels (t0 and t1) and “condition” with 6 levels (2F0B, 3F0B, 4F0B, 2F1B, 3F1B, 4F1B).

The variation of the remaining variables (MMHDs, VAS, MDMs, MIDAS, HIT-6) were studied using a matched-paired Wilcoxon signed rank test.

## Figures and Tables

**Figure 1 toxins-15-00023-f001:**
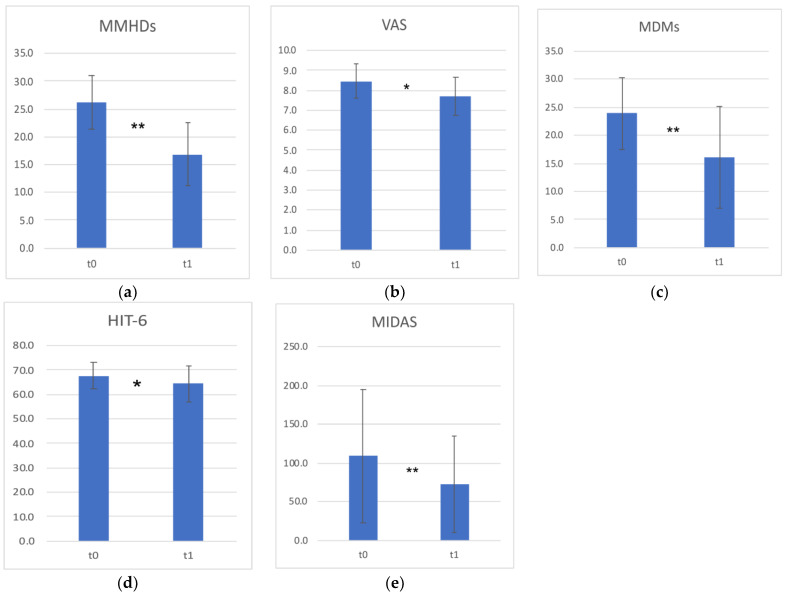
The mean clinical findings and questionnaires scores are represented: (**a**) Mean Monthly Headache Days, (**b**) pain intensity evaluated with the Visual Analogue Scale, (**c**) Mean number of Days with Medications, (**d**) Headache Impact Test-6, (**e**) Migraine Disability Assessment; vertical bars represent standard deviations; t0 = baseline, t1 = post-treatment; * *p* < 0.05; ** *p* < 0.01.

**Figure 2 toxins-15-00023-f002:**
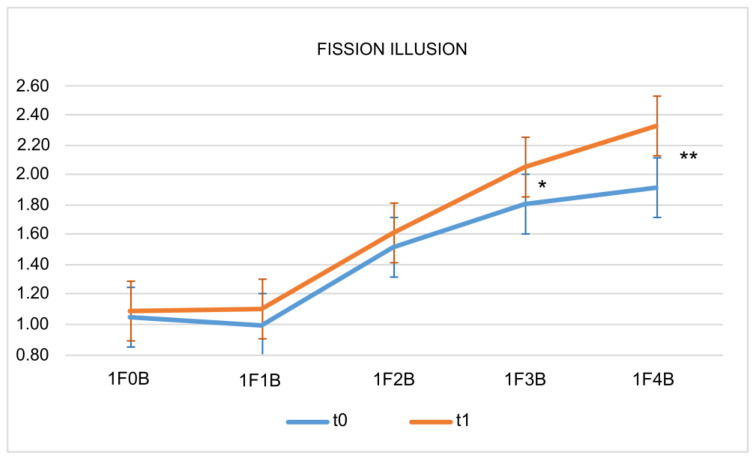
The mean perceived flashes’ variation between t0 and t1 regarding the fission illusion is represented. Bars = standard deviations; * *p* < 0.05; ** *p* < 0.01.

**Figure 3 toxins-15-00023-f003:**
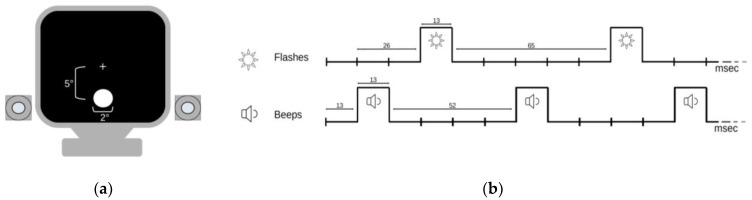
Schematic representation of SIFI procedure: (**a**) monitor presenting flashes with or without beeps; (**b**) timing of stimuli presentation, with numbers expressing the intervals in milliseconds (ms).

**Table 1 toxins-15-00023-t001:** Patients’ clinical characteristics and mean questionnaires scores variation between t0 and t1.

Variable	t0	t1	Significance *
MMHDs	26.2 ± 4.7	16.9 ± 5.6	<0.001
VAS	8.5 ± 0.9	7.7 ± 0.9	0.022
MDMs	23.9 ± 6.5	16.1 ± 9.0	0.001
HIT-6	67.8 ± 5.3	64.4 ± 7.4	0.018
MIDAS	108.9 ± 85.7	73.2 ± 62.2	0.004

MMHDs: Mean Monthly Headache Days; VAS: Visual Analogue Scale; MDMs: Mean number of Days with (acute) Medications; HIT-6: Headache Impact Test-6; MIDAS: Migraine Disability Assessment; t0 = baseline, t1 = post-treatment; * matched-paired Wilcoxon signed rank test.

**Table 2 toxins-15-00023-t002:** Mean number of flashes reported during fission trials at t0 (baseline) and t1 (post-treatment).

Condition	t0	t1	*p*-Value *
1F0B	1.0 ± 0.1	1.1 ± 0.2	0.649
1F1B	1.0 ± 0.0	1.1 ± 0.2	0.310
1F2B	1.5 ± 0.3	1.6 ± 0.4	0.329
1F3B	1.8 ± 0.5	2.1 ± 0.4	0.011
1F4B	1.9 ± 0.7	2.3 ± 0.5	<0.001

* post-hoc Duncan test.

## Data Availability

Data will be available after reasonable request to the corresponding author.

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
