# Peer review of "OnabotulinumtoxinA Modulates Visual Cortical Excitability in Chronic Migraine: Effects of 12-Week Treatment"

_toxins, 2022, doi:10.3390/toxins15010023_

Round 1

Reviewer 2 Report

Manuscript entitled „ Onabotulinumtoxin A modulates visual cortical excitability in chronic migraine: effects of 12 weeks treatment” is an interesting, well-written and well-planned experimental work. However, the text needs the little corrections according to the following comments:

Introduction

Line 73 – should be written allodynia

Line 82 – write [18]

Results

Line 114 Table 1 – please explain all the abbreviations in full name

Materials and Methods

Line 234 – use only abbreviation MMHDs

Line 238 - use only abbreviation VAS

Line 239 - use only abbreviation MDMs

Reviewer 3 Report

The Authors evaluated the effect of a single administration of BoNT-A therapy as prevention in chronic migraine and focused their attention on a possible neurophysiological marker: change in visual cortical excitability after 12 weeks. The manuscript is clear and presented in a well-structured manner. The cited references are relevant. The experimental design is suitable for testing the hypothesis but, as correctly highlighted by the Authors, a control group would have improved it. The sample included in the study is limited but the Authors highlighted this aspect in the discussion. The manuscript’s results could be reproducible based on the details given in the methods section. The figures and tables are appropriate. The conclusions are consistent with the evidence presented if limitations are taken into account. Ethical statements are adequate.

Minor spell check: line 194 "can be effectively used"

Round 2

Reviewer 1 Report

The authors have taken into account my suggestions